# The Lung Health Ambassador Program: A Community-Engagement Initiative Focusing on Pulmonary-Related Health Issues and Disparities Regarding Tobacco Use

**DOI:** 10.3390/ijerph18010005

**Published:** 2020-12-22

**Authors:** Panagis Galiatsatos, Eliza Judge, Rachelle Koehl, Marcella Hill, Olivia Veira, Nadia Hansel, Michelle Eakin, Meredith McCormack

**Affiliations:** Department of Medicine, Division of Pulmonary and Critical Care Medicine, Johns Hopkins School of Medicine, Baltimore, MD 21224, USA; ejudge2@jhu.edu (E.J.); rkoehl1@jhmi.edu (R.K.); mhill55@jhmi.edu (M.H.); oveira1@jhu.edu (O.V.); nhansel1@jhmi.edu (N.H.); meakin1@jhmi.edu (M.E.); mmccor16@jhmi.edu (M.M.)

**Keywords:** tobacco prevention, community engagement, school-based curriculum

## Abstract

*Introduction:* Educational campaigns have the potential to inform at risk populations about key issues relevant to lung health and to facilitate active engagement promoting healthy behaviors and risk prevention. We developed a community-based educational campaign called the Lung Health Ambassador Program (LHAP) with a goal of engaging youth and empowering them to be advocates for pulmonary health in their community. *Objective:* To evaluate the process outcomes and feasibility of the inaugural LHAP (2018–2019 academic year), with a specific aim to impact tobacco policy in the state of Maryland. *Methods:* Outcomes regarding feasibility included assessment of number of schools reached, number of students and healthcare professionals participating, and types of projects developed by participating students to impact modifiable risk factors for lung health. The courses for the LHAP were five 1 h sessions implemented at days and times identified by the community. The topics of the LHAP focused on lung anatomy, pulmonary diseases affecting school aged youth, tobacco use and prevalence, and air pollution (both indoor and outdoor). The fifth class discussed ways in which the students could impact lung health (e.g., policy and advocacy) and mitigate pulmonary disparities. *Main Results:* The LHAP was implemented at two elementary/middle schools, one high school, and two recreation centers within an urban metropolitan region. A total of 268 youths participated in the LHAP (age ranging from 11 to 18), whereby 72 (26.9%) were Hispanic/Latino and 110 (41.0%) were African American. Of the participating students, 240 wrote letters to local politicians to advocate for policies that would raise the legal age of acquiring tobacco products to 21. As for healthcare professionals, 18 academic faculty members participated in implementing the LHAP: 8 physicians and faculty staff and 10 nurses. *Conclusions:* The LHAP is a community-based program that provides education and training in advocacy with a goal of teaching about and, ultimately, reducing respiratory health disparities. The results from the first year demonstrate that the program is feasible, with success demonstrated in completing educational modules and engaging students. Next steps will include strategies to ensure sustainability and scalability to increase the reach of this program.

## 1. Introduction

Tobacco dependence is a leading cause of preventable morbidity and premature mortality in the United States, and its current use has resulted in health disparities for persons of certain race and socioeconomic status [1,2]. In the United States, close to 90% of current smokers began before the age of 18 years [3], raising concern in that early age of smoking initiation is associated with decreased rates of successful smoking cessation [4]. Therefore, aiming to reduce youth tobacco use would be a key strategy to curb future use of active smoking, a strategy often emphasized by the US Surgeon General for example [5], with probable resulting health consequences in reduction in tobacco-related premature mortality and health-related disparities.

Increasing the minimum age to purchase tobacco products and proper enforcement of such a policy has been shown to reduce tobacco use among youth and adolescents [6,7]. Growing support across the United States for raising the minimum legal age to purchase tobacco products to 21 years (Tobacco 21) has been seen in recent years [3,8,9,10]. For states where the age has increased to 21, such an increase in age has been shown to have significant public health consequences; for instance, raising the minimum age to purchase tobacco from 18 years to 21 years has shown to result in a 12% reduction in overall smoking rates [11]. Further, combatting tobacco dependence in Baltimore City is vital, where prevalence of smoking is greater than the US national average, and tobacco store outlets are easily accessible in areas of the City with high rates of minors [12,13].

School-based health education programs have been identified as cost-effective and efficient intervention strategies to help deliver insight into public health messaging [14]. Such programs can attenuate global health concerns, such as obesity [15] and mental health [16], in a cost-effective, multi-disciplinary manner. To gain the support of Tobacco 21 for the state of Maryland, a school-based health education program was created, called the Lung Health Ambassador Program (LHAP). The goal of the LHAP was educating and empowering youth to be advocates for lung health for themselves and their community, with a specific culmination in supporting Tobacco 21 for the state of Maryland.

The objective of this study is to evaluate the process outcomes and feasibility of a multi-disciplinary, community based, health education program directed at youth. Further, we describe the various outcomes of the LHAP directed in support of Tobacco 21 for the state of Maryland.

## 2. Methods

### 2.1. Community Organizations

Schools and recreation centers were identified if they were within Baltimore City or Baltimore County. The schools were made aware of our initiative through a network of community organizations organized by Johns Hopkins School of Medicine’s Medicine for the Greater Good (MGG) [17]. Through MGG’s network, the LHAP was broadcasted, and community partners (e.g., schools, recreation centers) could then volunteer to have the LHAP implemented to their community. Classes selected ranged from 7th grade to 12th grade. For schools, when the LHAP would be taught, during school hours versus after-school hours, was decided at the discretion of the school. If the LHAP was decided to occur during school hours, it was implemented into a time frame usually designated for a science course, and students participating were present as part of their mandatory academic schedule. These time slots ranged from 45 to 55 min. If the LHAP was decided to be held after school, students would participate on a voluntary basis. The time slot for after school was 60 min. Each participating school for the LHAP had a member present along with the LHAP staff (e.g., science teacher).

For recreation centers, the LHAP was added as part of the after-school programs that were offered to neighborhood youth and adolescent. Participating in the LHAP was voluntary for the students; however, the age range allowed was for 12 to 18 years old, correlating with the 7th to 12th grade requirement for the schools. The time allotted for the LHAP at the recreation centers was 60 min. Recreation centers had to designate a team member for each LHAP class as well (e.g., a counselor).

Contextual-level variables were collected indicating where the communities were located at the census tract. Prevalence of smoking in adults, median income, and the area deprivation index (a census tract socioeconomic composite score ranging from 1 to 100, where the greater the score, the more disadvantaged the census tract) were determined based on Baltimore City Health Department data and US Census data [18,19].

### 2.2. Applying the Intervention

The topics of the LHAP individual classes are listed in Table 1. Each topic was intended to continue to build upon prior knowledge in order to emphasize the impact of modifiable variables that result in morbidity: tobacco use and exposure and environmental pollution. These topics were discussed in the context of not only contributing to pulmonary-related morbidity but also how they contribute to pulmonary-related health disparities. The health disparity component was emphasized throughout the entire LHAP curriculum, as it was meant to assure the advocacy objective of the fifth class would be executed in an effort to curb these disparities. The frequency of the classes was decided on by the schools, but ranged from once a week to devoting an entire week to the LHAP.

The topics of the LHAP were identified by physicians and faculty members of our research group (BREATHE Center). The pedagogy was approved by two middle school science teachers from Baltimore City. They emphasized the aforementioned time restrictions as well as hands on activities to maintain interest and help reaffirm the objectives. Further, the teachers emphasized having several volunteers in addition to the presenter, whereby the volunteers assisted with the hands-on activities by overseeing the students once they were divided into small groups.

Each class of the LHAP followed a similar work-flow. An introduction and recap of prior knowledge constituted 5% of the time, followed by formally teaching the material (40% of the time), then a hands-on activity to reaffirm the current knowledge taught (40% of the time), and closing with the students writing a reflection on what they have learned (10% of the time) and a preview for the next class (5% of the time). All staff from the LHAP, including the community organization members, helped with the teaching of the material as well as assuring the students followed the instructions for the hands-on activities.

The advocacy class (the fifth class) allowed for an overview of the LHAP, culminating then with the majority of the time discussing advocacy work. To assure Maryland’s Tobacco 21 would become law, the students were taught how to write letters to local and state politicians, as well as create media and flyers to disseminate within their community organization. Finally, if students desired, they were taught how to present testimony at hearings for Tobacco 21 that would be conducted in Annapolis, Maryland. The graduation ceremony for the LHAP occurred at the end of the academic year, regardless of when the LHAP was implemented, and would draw the entire audience of the community organization in an effort to highlight the great work of the students.

### 2.3. Program Evaluation Methods

We monitored how many students participated in the LHAP at each site. Each session culminated with a hands-on activity (Table 1) and allowance for the last class to write a reflection piece on their experience and thoughts on what they had learned. The writing was open-ended and voluntary, where teachers emphasized that the students could note anything they felt was interesting or concerning from the teaching. When available, these reflection pieces were collected and reviewed. Next, LHAP student participants participated in an initiative regarding Tobacco 21 for the state of Maryland. Such projects were recorded in an attempt to monitor type of project, target audience (for instance, students or parents, school-based or neighborhood-based), and frequency of the project. Finally, we collected teacher reflections on the LHAP curriculum that were sent to us by email.

## 3. Results

The Lung Health Ambassador Program was implemented at four locations in Baltimore City: two elementary/middle schools and two recreation centers (Table 2). One community organization was a high school located in Baltimore County. The smoking prevalence of adults in the neighborhoods for the community organizations ranged from 26.2% to 30.2% (Table 2). Further, the median income of the neighborhoods the schools and recreation centers were located in ranged from USD 25,738 to 41,219. These areas of Baltimore City include the Armistead neighborhood, Curtis Bay neighborhood, Downtown Baltimore City, and the Highlandtown neighborhood.

At the elementary/middle schools, 229 students participated in total, while at the one high school, 23 students participated. In each of the three schools, the grade level’s respective science teacher participated in the development of the curriculum and its in-class implementation. Sixty students (34 from one elementary/middle school and 26 at the other) were of Hispanic/Latino ethnicity, and 102 were African American. At one of the elementary/middle schools, the English teacher participated in an effort to help students construct “advocacy letters” that would be sent to state-level politicians. Note that the curriculum was also translated into Spanish for 14 students, and a designated LHAP faculty would lead the course to the students in Spanish simultaneously.

At the recreation centers, 31 students participated in total. Twelve of the 31 students were of Hispanic/Latino ethnicity and 8 were African American. Youth staff leaders overseeing the after-school programs participated with LHAP faculty members to implement the curriculum.

Each class culminated with a hands-on activity as well as time to write a brief reflection piece on what the students had learned. Reflection pieces are captured in Table 3 with representative quotes, categorized by the LHAP topic they were recorded in. Further, each teacher and recreation center leader had the opportunity to provide their own feedback at the end of the LHAP, which is also captured in Table 3. A common finding was the emphasis on “disparities” and how it is impactful for their classmates and community.

Of the students participating in the LHAP through the recreation center, their project outcome focused on creating a smoke-free environment for the property around the building. Both participating recreation centers have playgrounds and courts for basketball surrounding the building. The students created signs to advocate for smoke-free outdoor areas surrounding the recreation centers. The signs were created with the supervision of the recreation center staff leaders. The participating students requested all of their signs to be displayed beginning in June, where the number of students and families participating in the recreation center increases due to summer enrollment.

As for the students participating at the local schools, 240 wrote letters to local politicians to advocate for policies that would raise the legal age of acquiring tobacco products to 21: 229 from the elementary/middle schools wrote letters (identified as mandatory by the teachers) and 11 from the participating high school (where writing letters was voluntary). Forty-five of the letters were specifically sent to the state’s governor. Further, three high school students participated in providing testimony in Annapolis, Maryland, on 27 February 2019 in an effort to show support from Baltimore City youths in favor of Tobacco 21.

## 4. Discussion

In this study, we demonstrated a community-based initiative, the Lung Health Ambassador Program, which is able to provide feasible education and training in advocacy for youth, with a specific aim targeting education about the harms of tobacco use and prevention. In the multidisciplinary approach of the LHAP, the students learned not just objective scientific considerations regarding the impact of tobacco use on lung health and indoor air pollution but how these findings have culminated in health disparities, especially within their local communities. Further, student participants showed great efforts to support the legislative action, Tobacco 21 for the state of Maryland, as this aligned with the educational goals of the LHAP, while being provided with outlets to implement meaningful support and advocacy.

Translating science into community action in an effort to promote health and prevent disease should be a priority for clinical research aiming to maintain relevance within the communities such research is conducted [20,21]. Such organized community engagement by scientists and researchers helps assure that the public has the proper information provided at appropriate science literacy levels with resulting pre-identified actionable community projects and advocacy. Many anti-tobacco campaigns exist that aim at youth prevention of smoking, from the Truth Campaign to various digital media platforms [22,23,24]. For instance, Ramirez-Andreotta et al. demonstrated how a community’s understanding of environmental health issues can change with information sharing and engagement with scientists and environmental health agents, allowing community health issues to be addressed at the local and state-level legislation [25]. The LHAP followed a similar platform to Ramirez-Andreotta et al., where research findings were discussed in an effort to raise concerns (disparities around tobacco and air pollution) while culminating in the ability to promote advocacy and action for the community members. This allows the LHAP to be in accordance with and support the mission of many other impactful campaigns around youth smoking prevention [23,24,26,27], while at the same time attempting to reaffirm the impact of in-classroom, in-person messaging to youth in school.

Research organizations, hospitals, and policymakers interested in enacting regulations based on the scientific and medical findings can use a model such as the LHAP to aid in endorsing concepts and actionable projects to promote health and prevent disease. Our data suggest that increasing awareness of the health disparities caused by the use of tobacco as well as secondhand smoke resulting in indoor air pollution was s influential in how the youth viewed this risk factor. Discussions of health disparities may be more influential in their ability to result in advocacy and action—as compared to traditional teachings of cause-and-effect regarding tobacco use and health outcomes—warranting this perspective be strongly considered in future public health discussions.

Several factors limit the generalizability of the findings of this study. First, several students did not participate in advocacy projects at the end of the LHAP. Exploring why students may not take part in the final request of the LHAP is warranted for moving forward and assuring the benefit of all students. Second, the students participating in the LHAP were of an urban metropolitan region. It is unclear if schools and communities in rural areas would gravitate to insight into pulmonary-related disparities in risk factors and health outcomes. Third, it would be ideal to receive qualitative feedback from teachers and parents regarding the LHAP. While this portion of the pilot focused on feasibility and specific process outcomes, moving forward, gaining insight into such feedback from the community will be vital to reaffirm its significance in schools and communities alike. Finally, whether such a curriculum can adapt to future issues concerning pulmonary health and outcomes is unclear. Investigations into whether the LHAP can provide opportunities to curb issues, such as the youth epidemic in electronic cigarette use, or to promote advocacy, such as around indoor air quality standards for housing units, remains to be explored.

The Lung Health Ambassador Program is a community-based curriculum that provides education and training in advocacy with a goal of reducing pulmonary health disparities. The efforts were executed in a multi-disciplinary manner with community and school leaders as well as healthcare professionals and scientists. The results from the first year demonstrate that the program is feasible, and factors have been identified that warrant further exploration in an effort to ensure the program’s generalizability. Next steps include strategies to reach more diverse communities (e.g., rural areas) and attempt to identify the role LHAP can play in other respiratory-related health issues.

## Figures and Tables

**Table 1 ijerph-18-00005-t001:** Summary of the Lung Health Ambassador Program courses, objectives, and hands-on activities.

Topic:	Objective(s) and Activity:
Lung Anatomy and Function	To understand the basic function and anatomy of the lungs. ○ How oxygen gets into the lungs and carbon dioxide out? ○ Structure of the lungs: airways, lung muscles, and circulation system of the lung. ○ How doctors examine lungs to assure they are healthy. Hands-On: To build model lungs using arts and crafts.
Asthma	To understand what asthma is and how to manage the disease. ○ What is asthma? ○ What are triggers of asthma? ○ How to manage asthma. Hands-On: To build model lungs that demonstrate how it feels to breathe when a person has an asthma attack.
Environmental Air Pollution	To explore what is air pollution and how students can impact the amount of pollution that occurs in their homes and local community. ○ What is pollution? ○ How does tobacco smoke contribute to indoor air pollution? ○ What is the purpose of an air filter? ○ How does air pollution worsen certain lung diseases, such as asthma and COPD? Hands-On: To build a model air filter and understand how home air filters work.
Smoking and Tobacco	To engage the students in understanding how smoking and secondhand smoke impacts lungs, especially those with asthma. ○ What makes cigarettes (combustibles and electronic) harmful? ○ What is secondhand and thirdhand smoke? ○ What happens to the lungs when they breathe in smoke? ○ How does smoking impact the lungs and lung diseases, such as asthma and COPD? Hands-On: To create a plan to reduce the impact of peer pressure on smoking.
Ambassador Training	This will serve as a recap of the prior classes in order to create action plans for targeting key objectives for the students to take on over the summer months. Sessions on advocacy run in collaboration with local affiliates.
Graduation	Certificates and a ceremony to welcome the Lung Health Ambassadors.

**Table 2 ijerph-18-00005-t002:** Summary of the participating schools and recreation centers. Where applicable, data are provided at the census track level.

Community Partner	Participants	Percentage of Adult Smokers (%) *	Area Deprivation Index (National Score)
Elementary/Middle School 1	Classes: 7th and 8th Grade Total students: 118 Hispanic/Latino: 26 African American: 72 Caucasian/White: 20 Curriculum implemented during school hours	26.2	88
Recreation Center 1	Classes: 7th to 9th Grade Total students: 14 Hispanic/Latino: 5 African American: 2 Caucasian/White: 7 Curriculum implemented during after school hours	30.1	90
Recreation Center 2	Classes: 6th to 8th Grade Total students: 17 Hispanic/Latino: 7 African American: 2 Caucasian/White: 8 Curriculum implemented during after school hours	24.7	84
Elementary/Middle School 2	Classes: 7th and 8th Grade Total students: 111 Hispanic/Latino: 34 African American: 30 Caucasian/White: 47 Curriculum implemented during school hours	30.2	98
High School	Classes: 10th to 12th Grade Total Students: 23 Hispanic/Latino: 0 African American: 4 Caucasian/White: 19 Curriculum implemented during after school hours	Not Available	72

* Provided by the Baltimore City Health Department. Note that the High School resides in Baltimore County and adult smoking prevalence is not readily available.

**Table 3 ijerph-18-00005-t003:** Themes from participants of the Lung Health Ambassador Program (LHAP) as captured during individual classes.

Key Theme	Representative Quote
Lung Anatomy and Function	“Lungs are made to get oxygen in” “Lungs are still growing at my age” “Harming lungs now will keep them from reaching their potential”
Asthma	“Asthma is common in my class, and that’s not fair” “Lots of things can make asthma bad. I see a lot of them in my community”
Environmental Air Pollution	“Lots of people smoke in my community. Secondhand smoke is bad for me and my friends to breathe.” “Some of my friends talk about smoking. If they wanted to, I think they can get cigarettes.” “It’s easy to get cigarettes in Baltimore.”
Smoking and Tobacco	“Air pollution is also about air in my home.” “Secondhand smoke is a type of air pollution.”
Ambassador Training	“I can make a change now. I can help make laws to protect my friends from having lung diseases.”
	“It bothers me how being poor means you may smoke more. I want to protect my community.” “I believe I can make a difference. Disparities shouldn’t exist around smoking and pollution.”
	“The attractiveness of the Ambassador Program is its focus on disparities that effect these students, their families, and their community. I believe it motivated them greatly to act in some manner to help assure these disparities are not present in the future.” ^Τ^
	“Having the ability to interact with nurses, doctors, and scientists is a great benefit to the program. The students began to identify themselves with each instructor.” ^Τ^

^Τ^ Quotes taken from teachers and recreation center leaders.

## Data Availability

The data presented in this study are available on request from the corresponding author.

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
