# Peer review of "The Lung Health Ambassador Program: A Community-Engagement Initiative Focusing on Pulmonary-Related Health Issues and Disparities Regarding Tobacco Use"

_ijerph, 2020, doi:10.3390/ijerph18010005_

Round 1
Reviewer 1 Report
Youth tobacco use, particularly in low-income and ethnic minority communities, remains a significant public health problem. The LHAP promises to educate, engage and empower youth to address this issue in their communities, and hopefully refrain from tobacco use on their own. The paper represents an important contribution to the literature in sharing the details of the first phases of implementing the program.
There are a number of issues, outlined below, that should be addressed to strengthen and polish the paper prior to publication.
Abstract notes that five sessions were implemented over days at times identified by the community. Either number of days should be included or perhaps it should state at days and times identified by the community?
- Consider using youth rather than children to describe the population that spans up to age 18.
Introduction notes that aiming to reduce youth tobacco use would be a key strategy to curb future use of active smoking.... This reads as though the authors are proposing this as a somewhat novel idea. The US Surgeon General has been calling for this for many years, along with other federal agencies. This should be noted. The descriptions of policy and school-based approaches that follow are helpful, but noting the widespread consensus around the need for a youth-focused approach up front would help with the framing.
- It would be helpful to provide some context to explain the specific need for this work in Baltimore. Disparities in use are mentioned generally, but what are the rates in Baltimore, generally and in the specific population groups that were targeted in this work? How do they compare with national rates? Generally explain to the reader why this program is needed in this particular community.
- Page 2, line 20 contains a typo: perhaps should read youth directed program?
Methods
- Additional information is needed about how the schools and rec. centers were identified and recruited. Not all Baltimore City and/or County schools were included. How were sites chosen to be invited? How were sites selected among from those that expressed interest. Same for community centers.
- Reference is made in Line 40 of Page 2 to "data collected" that actually came from the local health department and the census. This is a bit misleading. More appropriate to say something like: Prevalence of smoking, median income, etc... were determined based on data from the Baltimore City Health Department and US Census.
- Page 2, Line 37: contribute may not be the best word, consider designate
- Page 2, Line 39: consider replacing 'of where' with 'indicating where'
- Page 3, Line 3 has a missing word: should be decide on BY the schools
- First full para on page 3 -- stick with past or present tense vs. flipping back and forth (same comment for throughout the manuscript)
- Table 1 on Page 3 is very informative. A few places where punctuation is missing.
Results
- Table 2 includes an area deprivation index, but I could not find this metric referenced at all in the text. What is it telling the reader?
- Table 3 does an excellent job of showing student (and teacher) perspectives and adds an important level of richness to this section of the paper. If possible, it would be great to include a figure/s and/or additional table to show examples of posters made by some of the students as well as quotes from some of the letters to politicians and/or testimony given by the students.
- A table that includes demographic characteristics of the participants, perhaps separated by school vs. rec. center, would be very helpful. At present, race/ethnicity of some of the participants are listed, but it is left to the reader to determine the proportion of the sample that fall into these categories.
- Additional information about which students participated in which activities would be helpful to the reader, particularly some description of those that completed the advocacy activity vs. those who did not.
- Were there any differences between the groups of students who were "required" to participate as a part of their science class vs. students who voluntarily participated in an afterschool program?
- Page 4, Line 16: should state where the schools and recreation centers are located vs. reside
Discussion
There are a number of long-standing public health campaigns that call attention to disparities in tobacco use rates in relation to ethnicity and socio-economic status. Examples include the Truth Campaign, Stops with Me, and Tobacco Free Kids. These should be referenced in some way in the 2nd para on Page 6. At present, the para reads as though this is a novel approach being suggested by the authors.
- Page 6, Line 18: strongly consider rephrasing "increasing public awareness to youth"
- Page 6, Lines 37-38: typos here need to be corrected
Author Response
Please see our Cover letter that addresses Reviewer 1 and Reviewer 2.

Reviewer 2 Report
I think that process evaluation studies are very much needed. This is one such study, and it does a good beginning job of process evaluation. The limit here is that the data collected are pretty limited. Participation, content of lessons and then student reflections. For me, more is needed about the challenges of the implementation to really understand if this intervention is meaningful. Interviews with the instructors, feedback from students and parents are really important data here that I think are missing in this process evaluation.
Author Response

(The authors gave the same response as above.)
